# Costs of Providing HIV Self-Test Kits to Pregnant Women Living with HIV for Secondary Distribution to Male Partners in Uganda

**DOI:** 10.3390/diagnostics10050318

**Published:** 2020-05-19

**Authors:** Michelle A. Bulterys, Andrew Mujugira, Agnes Nakyanzi, Miriam Nampala, Geoffrey Taasi, Connie Celum, Monisha Sharma

**Affiliations:** 1International Clinical Research Center, Department of Global Health, University of Washington, Seattle, WA 98104, USA; mbult@uw.edu (M.A.B.); mujugira@uw.edu (A.M.); ccelum@uw.edu (C.C.); 2Infectious Diseases Institute, Makerere University, Kampala P.O. Box 22418, Uganda; anakyanzi@idi.co.ug (A.N.); mnampala@idi.co.ug (M.N.); 3Division of HIV Testing, Uganda Ministry of Health, Kampala P.O. Box 7272, Uganda; taasi.taasi@gmail.com

**Keywords:** HIV testing, HIV self-testing, secondary distribution, costing, pregnant women, Africa

## Abstract

Background: Secondary distribution of HIV self-testing kits (HIVST) to pregnant women attending antenatal care (ANC) clinics to give to their male partners is a promising strategy to increase testing coverage among men, but its costs are unknown. Methods: We conducted micro-costing of a trial evaluating secondary distribution of HIVST on pregnant women living with HIV (PWLHIV) in an ANC in Kampala, Uganda. Costs (2019 USD) were collected from program budgets, expenditure records, time and motion observations, and staff interviews and estimated for three scenarios: as-studied, reflecting full costs of the research intervention, Ministry of Health (MOH) implementation, reflecting the research intervention if implemented by the MOH, and MOH roll-out, the current strategy being used to roll out HIVST distribution. Results: In the as-studied scenario, cost of HIVST provision was $13.96/PWLHIV reached, and $11.89 and $10.55 per HIV-positive and HIV-negative male partner, respectively, who linked to a clinic for facility-based testing. In the MOH implementation scenario, costs were $9.45/PWLHIV, and $7.87 and $6.99, respectively, per HIV-positive and HIV-negative male partner linking to the clinic. In the MOH roll-out scenario, the cost of HIVST provision to pregnant women regardless of HIV status was $3.70/woman, and $6.65/HIV-positive male partner. Conclusion: Secondary distribution of HIVST from pregnant women can be implemented at reasonable cost to increase testing among men in Uganda and similar settings in Africa.

## 1. Introduction

HIV testing is the critical entry point to accessing lifesaving antiretroviral treatment (ART) and prevention strategies. However, the potential of ART to curb the HIV epidemic in sub-Saharan Africa (SSA) is hindered by the gender disparity in HIV testing uptake. Compared to women, men in SSA have lower HIV testing rates and link to ART later in the course of their illness at lower CD4 counts, resulting in poorer clinical outcomes and increased transmission to female partners [1]. Studies highlight men’s preference to test outside of facilities [2] to avoid testing barriers including travel distance, long wait times, costs (transport and lost wages), confidentiality concerns, and cultural beliefs that clinics provide services for women and children [3,4,5,6]. Community-based HIV testing (testing outside of clinic settings) has the potential to overcome barriers associated with facility-based HIV testing and achieve high coverage among men in SSA. 

One community-based HIV testing strategy with high acceptability among men is HIV self-testing (HIVST) [7]. HIVST is a convenient and discrete alternative to facility-based testing that can overcome men’s barriers to testing, including time and transport costs, stigma, and confidentiality concerns. In a community-based trial in Malawi, HIVST achieved 76% testing coverage among men and was reported as the preferred option for future HIV testing [7]. A particularly promising HIVST delivery strategy is secondary distribution from females to their male partners. In this model, women in antenatal clinics (ANCs) are given an HIVST by a healthcare provider and trained to use the HIVST, interpret results, and deliver it to her male partner. This strategy has been shown to increase HIV testing among men, couples testing, linkage, and mutual disclosure of HIV status [8,9,10]. In addition to increasing HIV testing among men, secondary distribution of HIVST can have health benefits for women regardless of their HIV status. HIV-negative pregnant women are at high-risk of acquiring HIV, partially due to low testing rates among their male partners; HIVST can increase men’s knowledge of their HIV status and facilitate linkage to care, which can reduce transmission to partners. For pregnant women living with HIV (PWLHIV), HIVST distribution can promote couples testing and disclosure, which can increase women’s retention in ART and engagement in PMTCT regimens, improving the health of women and infants [11]. Furthermore, testing partners of PWLHIV is a high yield strategy to identify HIV-positive men; studies find HIV positivity of 50%–60% in primary partners of HIV-positive individuals, many of whom are unaware of their status [12,13]. Male partners of PWLHIV who test HIV-negative will be in HIV sero-discordant partnerships and can benefit from linking to HIV prevention, including pre-exposure prophylaxis (PrEP). 

High fertility rates in SSA coupled with high antenatal care (ANC) attendance (98.4% in Uganda) result in the majority of women attending clinics for HIV testing in their lifetime; therefore, secondary distribution has the potential to reach a large proportion of the male population [14,15]. One study found that 91% of ANC attendees reported successfully distributing HIVST kits to their male partners, which facilitated couples testing and disclosure as well as safer sexual behaviors; however, most of the women included in this study were HIV negative so HIV status disclosure is not a barrier to delivery of HIVST to their partners [9]. In response to studies showing high testing uptake, the WHO issued guidelines recommending scale-up of HIVST distribution, including secondary distribution to pregnant women in order to close the testing gap among men. Several countries in SSA have begun national scale-up of secondary distribution of HIVST as part of routine care in ANC clinics. To our knowledge, the programmatic costs of this strategy have not been evaluated. Estimating the cost of secondary distribution of HIVST in ANCs is useful for budgetary planning for ministries of health, donors, and other stakeholders designing policies to scale-up HIVST. Costs can also be used to form economic evaluations including budget impact and cost-effectiveness analyses. Our research objective was to estimate the incremental cost of incorporating secondary distribution of HIVST into routine care in ANC clinics in Uganda, estimating costs for three scenarios of varying staff and resource intensity. This study is timely as the Uganda Ministry of Health began rolling out this intervention in ANC clinics across the country starting in January 2020.

## 2. Materials and Methods 

### 2.1. The Obumu Study

The present study was nested within an ongoing randomized clinical trial (Obumu) conducted at Kitebi Health Centre III, a public clinic in Kampala, Uganda (www.ClinicalTrials.gov NCT03484533). The primary goals of Obumu are to evaluate the impact of secondary distribution of HIVST from HIV-positive pregnant women on 1) male partner’s testing and linkage to ART or pre-exposure prophylaxis (PrEP), and 2) women’s post-partum ART continuation and viral suppression. Eligibility criteria include HIV-positive pregnant women ≥ 18 years accessing antenatal care at Kitebi Health Centre III, who have a male partner of unknown HIV status, and are at low risk of intimate partner violence. The majority of women enrolled in Obumu had not disclosed their HIV status to their male partners by the time of study enrollment (68.2%). Women (*N* = 500) were randomized 2:1 to receiving an HIVST or an invitation letter to their partner for fast-track testing at the antenatal clinic (standard of care). Women in the intervention arm receive individual counseling from a healthcare provider on the use and interpretation of HIVST and strategies to approach their partner to increase his HIVST uptake. All men are encouraged to come to the antenatal clinic for a provider-administered HIV test. Men who test HIV-positive are linked to HIV care and HIV-negative men are offered PrEP. Study recruitment was completed in February 2020, and women and male partners will be followed until 12 months post-delivery. 

### 2.2. Micro-Costing 

We conducted a micro-costing study at the Obumu study clinic to capture the economic costs of implementing secondary distribution of HIVST and male partner testing after clinic linkage. Costs (2019 USD) were collected using a provider perspective from expense reports, staff and expert interviews, and divided into: personnel, transportation, equipment, supplies, buildings and overhead, start-up, and phones/data monitoring. We developed an initial list of main activities through review of the project protocol and discussion with the site team. We conducted semi-structured interviews with study and facility staff to obtain information on resource use and staff time needed for HIVST distribution. Unit costs were estimated separately by HIV status for pregnant women and male partners.

### 2.3. Time and Motion Observation

We conducted time and motion observations over 4 weeks in August 2019 in the Obumu study clinic to estimate staff time needed to implement the intervention. When a study participant arrived at the clinic research room, a stopwatch was started and the amount of time each activity took was recorded using pen and paper (e.g., screening, informed consent process, HIV test, counseling, HIVST training, and research questionnaires). Time and motion observations were conducted until we reached information saturation, and data were extracted into Microsoft Excel spreadsheets to calculate total and unit costs associated with the intervention. Resources and time spent on research (e.g., administering informed consent and study questionnaires) and routine clinical activities (e.g., ART dispensing, adherence counseling, and viral load testing) were removed from programmatic costs. 

### 2.4. Scenarios and Assumptions

Time and motion observations and staff interviews were used to inform productivity assumptions. We estimated the daily average number of pregnant women that could be counseled on distributing HIVST to their male partners and the daily average number of male partners that could be provided confirmatory HIV testing after HIVST distribution (see Appendix A for details). We assumed the clinic was operating at full capacity and staff conducted routine standard of care activities if they were not working on the HIVST intervention. Capital costs (e.g., equipment, furniture) and start-up costs (e.g., staff hiring, training) were annualized assuming 5-year useful life and discounted annually at 3% [16]. We assumed that all male partners who linked to the clinic would receive an HIV and syphilis test as per Uganda national guidelines [17]. A positive HIV test result was associated with added costs of confirmatory testing and additional counseling. Intervention costs were estimated separately for pregnant women and male partners by HIV status and were calculated as the annual intervention costs divided by the number of participants reached per year. Assumptions on patient volume, staff structures, and all costing inputs are described in the Appendix A. 

We calculated programmatic costs for three intervention scenarios (Figure 1):

### 2.5. Scenario 1: As-Studied 

This scenario reflects the cost of implementing HIVST distribution as conducted in the Obumu intervention, whereby PWLHIV received individual counseling on HIVST use and strategies to encourage their male partners to use the HIVST kit. Individual counseling was provided in Obumu since all pregnant women were living with HIV. Women were given tools to prepare them for disclosing their HIV status if they desired, as well as how to provide some initial counseling to their partner regarding HIVST, and benefits of PrEP and ART depending on his status. A male health care provider calls male partners (with permission from pregnant women) and encourages them to link to the clinic for confirmatory testing regardless of the results of their HIVST. Further, men receive a transport reimbursement for attending the clinic. Personnel costs were calculated using the annual salaries of the research staff delivering the intervention, including a 34% benefits rate. Start-up costs included the development of standard operating procedures and a 3-day offsite training for all staff associated with intervention implementation. Clinical supplies costs included the HIVST kits provided to women, and rapid HIV and syphilis tests for all male partners who come in for a clinic visit (refer to Appendix A). 

### 2.6. Scenario 2: MOH Implementation 

This scenario reflects the estimated program costs of the Obumu intervention if implemented by the Uganda Ministry of Health (MOH). Similar to the as-studied scenario, we assumed PWLHIV received individual HIVST counseling but we used government health sector salaries for personnel costs instead of research staff salaries. We used government vendor-negotiated costs for clinical supplies instead of study costs, which were obtained from personal communications with policy makers at the Uganda Ministry of Health. To maintain consistency with the Obumu intervention, we assumed men receive a travel reimbursement upon linking to the clinic. 

### 2.7. Scenario 3: MOH Roll-Out 

This scenario reflects the program costs of secondary distribution of HIVST as currently being rolled out by the Ugandan MOH. The MOH is providing HIVST to all pregnant women who attend public ANC services, regardless of their HIV status. Instead of individual counseling as performed in the Obumu intervention for PWLHIV in order to assess risk of social harm if a woman discloses her HIV status and provide strategies on how to approach male partners with HIVST kits, clinic staff conduct group counseling sessions, with group sizes ranging from 20 to 70 depending on the facility size. Through time and motion observations at the Obumu study clinic, we estimated an average of 30 women attended the daily HIVST group counseling sessions, which are led by nurses and peer support counselors, and last for approximately one hour. The sessions consist of HIVST training and demonstration, counseling on strategies for discussing HIV self-testing with male partners, and instructions on how to encourage men who test positive to seek confirmatory testing. At the end of the session, women are offered an HIVST kit and asked to write their contact information into a MOH record book for follow-up and MOH monitoring and evaluation. We assumed that 80% of women in these MOH status-neutral group counseling sessions agreed to take an HIVST kit. While male partners in the previous two scenarios were encouraged to link to the clinic for confirmatory testing regardless of their HIVST result, with HIV-negative men being offered PrEP, we assumed that only men who self-test HIV-positive are encouraged to come to the clinic in the MOH scenario, reflecting current HIVST delivery guidelines in Uganda. Based on communications with staff at the MOH, we assumed initial provider training on HIVST distribution was conducted by two trainers from the MOH, who led one-time training sessions at designated facilities. Each facility training lasted a total of 1.5 days (12 h over 3 days), and involved at least 2 nurses/midwives, 4 peer counselors, a data manager, and a laboratory technician. Supervision of training, and program monitoring and evaluation were conducted by an MOH representative. 

### 2.8. Ethics Statement

The University of Washington Human Subjects Review Committee (IRB ID: STUDY00002257, 2 June 2017), National HIV/AIDS Research Committee (NARC 200, 28 July 2017) and Uganda National Council for Science and Technology (UNCST SS4501, 21 February 2018) approved the study protocol. All participants provided informed consent.

## 3. Results

We conducted 32 time and motion observations from July to August 2019, which included female enrollment and follow up visits, and male enrollment and follow up visits. The unit cost of HIVST kits was assumed to be $3.05 in all three scenarios. The costs of utilities and building, office supplies, and vehicles and maintenance were the same across the as-studied and MOH implementation scenarios (scenarios 1 and 2). 

### 3.1. Scenario 1: As-Studied 

The average programmatic cost of distributing an HIVST kit to a PWLHIV in scenario 1 was estimated to be $13.96 (Table 1).The cost of providing facility-based HIV testing and counseling to male partners who link to the clinic was $11.89 and $10.55, for men testing HIV-positive and HIV-negative, respectively (Table 2). Male linkage costs increased to $20.02 and $18.68, for men testing HIV-negative and HIV-positive, respectively, if an $8.13 travel reimbursement was given to males who linked to clinic testing, as done in the Obumu intervention. Overall, personnel comprised the largest proportion of the cost (73.4%), followed by HIVST kits (15.5%) (Figure 1).

### 3.2. Scenario 2: MOH Implementation 

The average cost of secondary distribution of HIVST kits by a PWLHIV, assuming the Obumu intervention was implemented by the MOH, was $9.45 (Table 1). The average cost providing facility-based HIV testing for men who link to the clinic after secondary distribution of HIVST was $7.87 and $6.99 per male partner who tests HIV-positive and HIV-negative, respectively. These costs increased to $16.00 and $15.12, respectively, if an $8.13 travel reimbursement was given to men to incentivize clinic linkage for HIV testing. Similar to the as-studied scenario, personnel comprised the largest proportion of the cost (60.6%), followed by HIVST kits (24.0%) (Figure 1).

### 3.3. Scenario 3: MOH Roll-Out 

In the status-neutral MOH scenario, personnel costs were considerably lower largely due to the assumed use of group rather than individual counseling. The average cost of distributing an HIVST kit to a pregnant woman, regardless of her HIV status, was $3.70 (Table 1). The average cost of clinic linkage for HIV testing for a male partner who self-tested positive was $6.65 per male (Table 2). We assumed men in this scenario were not provided travel reimbursements for testing at the clinic. HIVST kits comprised the largest proportion of the cost (75.5%), followed by personnel (12.8%) (Figure 1).

### 3.4. Overall Cost Proportions

Figure 2 presents the proportional breakdowns of the total cost for each scenario. Because Scenarios 1 and 2 involved individual counseling, personnel comprised the largest proportion of the cost (73.4% and 60.6%, respectively), followed by the cost of HIVST kits. In Scenario 3, group counseling allowed the personnel costs to drop, and therefore HIVST kits comprised the largest proportion of total costs. 

## 4. Discussion

We explored the relationship between costs of secondary distribution of HIVST kits by pregnant women to increase male partner testing uptake and linkage to care under varying intervention scenarios in Uganda. We found that secondary distribution of HIVST kits in ANC settings can be implemented at reasonable costs. Programmatic costs were highest in the as-studied scenario and decreased when assuming the intervention was implemented by the MOH (scenario 2). Costs were the lowest in the Ministry of Health scenario, which is currently being scaled-up in Uganda. Although the Ministry of Health scenario is the most applicable to the present HIVST distribution strategy in Uganda, we present the costs of more intensive interventions aimed at increasing uptake among HIV-positive pregnant women.

Personnel costs make up the majority of program costs in scenarios 1 and 2, which assumed individual HIVST counseling is provided to each pregnant woman. The national rollout of secondary distribution of HIVST kits in Uganda is utilizing group counseling for kit provision, significantly reducing the amount of time health providers spend administering the intervention. However, since PWLHIV may face greater consequences of delivering HIVST kits (e.g., accidental HIV disclosure, relationship dissolution, etc.), individualized counseling and risk-assessment, as provided in the Obumu trial, could increase the proportion of women who successfully deliver kits to their partners and encourage them to link to the clinic for HIV care or prevention. While this strategy is more costly than group counseling, cost-effectiveness analyses are needed to assess if it provides good value for money. Further, individualized counseling and risk assessment may lower risks of adverse events (e.g., intimate partner violence or relationship dissolution). If HIVST distribution is scaled up using group counseling, programs should be monitored for adverse events.

Similarly, the provision of a travel reimbursement for men upon linkage to the clinic can increase the proportion of men who attend the clinic after HIVST. Financial costs associated with clinic attendance (e.g., transport and lost wages) have been cited as barriers to men’s clinic linkage [3]. Providing a financial incentive in the form of a transport reimbursement can offset these costs and motivate men to come to the clinic for confirmatory testing, counseling, and linkage to HIV prevention services (e.g., PrEP) or HIV care. Further, incentives also provide a near term reward for a health behavior. Prior studies have shown the benefits of financial incentives to increase HIV testing [18], and clinic linkage [19]. A trial of incentivized mobile testing among men in Malawi found more first time testers and HIV-positive men in the incentivized compared to the non-incentivized mobile testing arm [20]. Future economic analyses should be conducted after the unblinding of the Obumu trial to assess testing coverage among men compared to the MOH roll-out scenario and determine if the added costs of individual counseling and travel reimbursements cost-effectively improve clinical outcomes.

While previous studies have estimated the costs of HIVST delivery strategies, to the best of our knowledge, this is the first micro-costing study of secondary distribution of HIVST. One study in Malawi conducted a costing of door-to-door HIVST to individuals in the community as part of a randomized clinical trial [9] and estimated the cost of HIVST distribution to be $8.78 per person; costs included HIVST provision and clinic-based confirmatory testing (USD 2014) [21,22]. This cost fell between our estimates for the as-studied and MOH roll-out scenarios. Another study assessing the cost of door-to-door community-based HIVST found the cost per individual given a kit was $8.15, $16.42, and $13.84 in Malawi, Zambia, and Zimbabwe, respectively, which are similar to our findings [23]. The majority of HIVST costing studies were conducted in the context of cost-effectiveness analyses, which found that HIVST is cost-effective when used for testing high-risk sub-populations in settings where the undiagnosed prevalence is above 3%, and if HIVST promotes linkage to PrEP and voluntary male medical circumcision (VMMC) [21]. A modeling study evaluating HIVST in low-income countries projected that HIVST would save $75 million USD in healthcare costs and avert up to 7000 disability-adjusted life-years (DALYs) over 20 years [24,25,26]. However, we did not find cost or cost-effectiveness studies on secondary distribution of HIVST from PWLHIV, limiting our ability to compare our results with others in the literature. 

Our analysis has several limitations. We conducted a micro-costing study within a clinical trial, yet our objective was to estimate programmatic costs of HIVST if implemented in a government setting. However, we conducted detailed time and motion observations and stakeholder interviews to remove research costs and include only costs relevant to programmatic implementation of secondary distribution. Further, the as-studied scenario only included estimated costs of HIVST provision for PWLHIV, since HIV-negative pregnant women were not eligible to enroll in the Obumu trial. However, it is likely that the intervention procedures would be similar for HIV-negative women and therefore the costs would generalize to pregnant women regardless of HIV status. Further, we conducted a micro-costing early in Uganda’s HIVST rollout when most participants were unfamiliar with HIVST, which posed a barrier for widespread uptake and trust in the test’s accuracy. As communities become more sensitized to HIVST, familiarity with the test will increase and counseling time may decrease. Our analysis only accounted for the costs of testing a male in the clinic, and did not account for the costs that are incurred after a man tests positive or negative (e.g., ART initiation and viral load testing if he tests positive, and creatinine testing and PrEP initiation if he tests negative). The as-studied scenario assumed that women only took home one kit despite being offered two; this is because women in the study were living with HIV and most had not yet disclosed their status to their male partner, and bringing two kits home could lead to her unintended disclosure. If women were to accept taking two kits home to test together, the costs would increase by the cost of another kit (USD $3.05). The as-studied scenario assumed that men received one HIV test and one syphilis test, and the MOH scenario assumed men received a dual rapid HIV/syphilis test per the Ugandan MOH guidelines. Despite one combined test costing less ($2 with subsidy) than two separate tests, there are perpetual stockouts and shortages of the dual test and therefore it is not always used. By assuming 100% usage of dual HIV/syphilis tests in the MOH scenario, we may be under-estimating the costs of providing syphilis screening as part of this intervention. Ongoing cost evaluations are needed to provide up-to-date cost evidence as HIVST continues to roll out to health facilities and public markets across the country. 

In summary, our costing analysis addresses the lack of primary cost data on HIVST distribution strategies which is useful as governments scale-up provision of HIVST kits as part of routine antenatal care. We found that secondary distribution of HIVST kits from pregnant women to their male partners can be implemented at reasonable costs to increase HIV testing coverage among men in Uganda. Our costs are likely generalizable to other countries in SSA with similar HIV epidemics. These results are timely as many countries in SSA, including Uganda, have recently begun rolling out secondary distribution of HIVST kits to pregnant woman in ANC to give to their male partners.

## Figures and Tables

**Figure 1 diagnostics-10-00318-f001:**
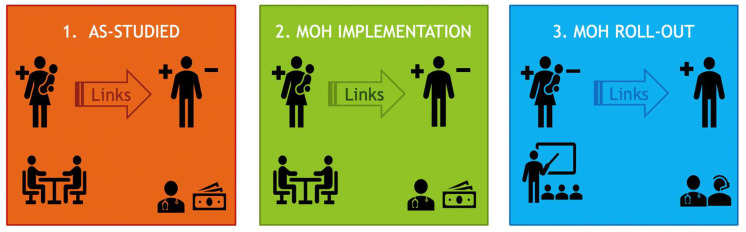
Intervention scenarios.*: The “as-studied” and “MOH implementation” scenarios both assume healthcare staff provides individual counseling with pregnant women living with HIV (PWLHIV) on HIV self-test (HIVST) use and strategies to give tests to their male partners. Regardless of whether males self-test HIV-positive or negative, they are encouraged to link to the clinic for confirmatory testing, counseling, linkage to treatment (if positive), PrEP (if negative), and syphilis testing; all men who attend the clinic receive a travel reimbursement. The “MOH roll-out” scenario assumes provision of group counseling for pregnant women, regardless of HIV status, on how to distribute HIVST kits to their male partners. Male partners are encouraged to link to the clinic only if they self-test positive; men attending the clinic receive confirmatory testing using the dual HIV/Syphilis rapid test, counseling, and linkage to treatment. Men in this scenario are not offered a travel reimbursement. The MOH currently offers toll-free phone counseling for anyone, male or female, interested in using an HIVST kit; therefore, this cost is included in the scenario.

**Figure 2 diagnostics-10-00318-f002:**
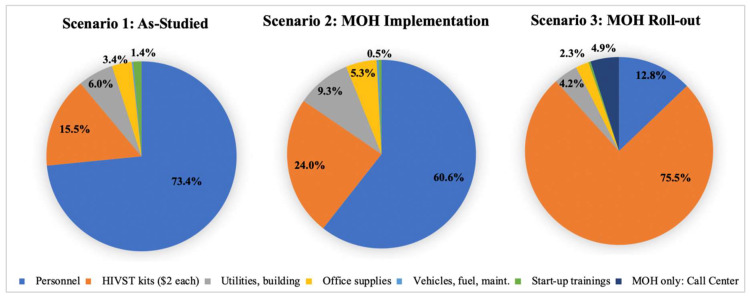
Allocation of HIVST delivery costs per female client by scenario.

**Table 1 diagnostics-10-00318-t001:** Average costs per HIVST kit distributed to a pregnant woman, by scenario (2019 USD).

	Scenario 1: As-Studied	Scenario 2: MOH Implementation	Scenario 3: MOH Roll-Out
HIV Status	HIV+ Females Only	HIV+ Females Only	HIV Status-Neutral
Counseling type	Individual	Individual	Group
Cost Category			
Personnel	9.48	5.06	0.34
HIVST kits	3.05	3.05	3.05
Utilities, building	0.78	0.78	0.11
Office supplies	0.44	0.44	0.06
Vehicles, fuel, maintenance	0.03	0.03	0.00
Start-up trainings	0.18	0.09	0.01
MOH only: Call Center	0.00	0.00	0.13
TOTAL COST/WOMAN	13.96	9.45	3.70

**Table 2 diagnostics-10-00318-t002:** Average costs per male tested for HIV in a clinic, as a result of secondary distribution of HIVST kits, per each scenario (2019 USD).

	Scenario 1: As-Studied		Scenario 2: MOH Implementation		Scenario 3: MOH Roll-Out
HIV Status	HIV+ Male	HIV- Male	HIV+ Male	HIV- Male	HIV+ Males Only
Cost Category					
Personnel	8.30	7.31	4.43	3.93	2.74
Clinical supplies for rapid HIV and Syphilis test	2.33	2.07	2.27	2.02	2.27
Utilities, building	0.68	0.61	0.68	0.61	0.91
Office supplies	0.39	0.34	0.39	0.34	0.51
Vehicles, fuel, maintenance	0.03	0.02	0.03	0.02	0.04
Start-up trainings	0.16	0.14	0.07	0.07	0.05
MOH only: Call Center	0.00	0.00	0.00	0.00	0.13
TOTAL COST/MAN	11.89	10.55	7.87	6.99	6.65
TOTAL COST/MAN + transport reimbursement ($8.13)	20.02	18.68	16.00	15.12	6.65 (no transport reimbursement)

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
