# Peer review of "Costs of Providing HIV Self-Test Kits to Pregnant Women Living with HIV for Secondary Distribution to Male Partners in Uganda"

_diagnostics, 2020, doi:10.3390/diagnostics10050318_

Round 1
Reviewer 1 Report
This manuscript presents costing for three different approaches to encourage HIV testing amongst men in Uganda. The first approach is being assessed in a randomised controlled trial that is currently underway. When results of the RCT are known/published, and the uptake rate of testing in men the researchers will be able to quickly undertake a cost benefit analyses.
Some specific points;
1) Consider adding an explicit research question at the end of the background. It is implied by "but its costs are unknown" but I think it would help the reader for the research question to be more clearly stated.
2) I know that this is not the subject of this manuscript but I was concerned when I read "low risk of intimate partner violence" Ln 90. Later on in the discussion it becomes clear that not all partners are aware of their pregnant partner/wife's HIV status. I'm really just making an observation and this isn't a review point, but those sections alerted me to the realities of conducting this research in this part of the world. You have my best wishes for success.
3) I thought that the reference or approval numbers for the various approvals or reviews should be provided in the Ethics Statement section.
4) Results section. It will assist the reader if you keep the words in the sub-headings the same as the tables and pie chart for consistency.
Author Response
Reviewer 1:
This manuscript presents costing for three different approaches to encourage HIV testing amongst men in Uganda. The first approach is being assessed in a randomised controlled trial that is currently underway. When results of the RCT are known/published, and the uptake rate of testing in men the researchers will be able to quickly undertake a cost benefit analyses. Some specific points;
1) Consider adding an explicit research question at the end of the background. It is implied by "but its costs are unknown" but I think it would help the reader for the research question to be more clearly stated.
Response: We thank the reviewer for this comment. The second to last sentence of our background states on Line 133: “We sought to estimate the incremental cost of incorporating secondary distribution of HIVST into routine care in ANC clinics in Uganda, estimating costs for three scenarios of varying staff and resource intensity.”
We have revised this to:
“Our research objective was to estimate the incremental cost of incorporating secondary distribution of HIVST into routine care in ANC clinics in Uganda, estimating costs for three scenarios of varying staff and resource intensity.”
2) I know that this is not the subject of this manuscript but I was concerned when I read "low risk of intimate partner violence" Ln 90. Later on in the discussion it becomes clear that not all partners are aware of their pregnant partner/wife's HIV status. I'm really just making an observation and this isn't a review point, but those sections alerted me to the realities of conducting this research in this part of the world. You have my best wishes for success.
Response: We thank the reviewer for this thoughtful comment. We agree this is an important aspect to highlight and we have added the following sentence to the methods section on Line 160 to make this point clearer earlier on in our paper:
“The majority of women enrolled in Obumu had not disclosed their HIV status to their male partners by the time of study enrollment (68.2%).”
In addition, we added the following to the discussion section on Line 353:
“Further, individualized counseling and risk assessment may lower risks of adverse events (e.g. intimate partner violence or relationship dissolution). If HIVST distribution is scaled up using group counseling, programs should be monitored for adverse events.”
3) I thought that the reference or approval numbers for the various approvals or reviews should be provided in the Ethics Statement section.
Response: We agree and have added all approval numbers to the Ethics Statement section.
4) Results section. It will assist the reader if you keep the words in the sub-headings the same as the tables and pie chart for consistency.
Response: We have now changed all sub-headings in the methods and results sections to be consistent with the tables and figures.
Reviewer 2 Report
The authors have conducted a great nested study within the Ibumu clinical trial. There have been other prevention publications studying HIVST distribution from PWLHIV to their partners but this is the first to address the costs perspective on distribution strategies. The manuscript was well written. There was good supporting evidence and rationale for the study and the study design was thorough and well thought-out. The results were presented in a concise manner and with clarity. Appreciate the authors addressing the study limitations. It would be worthwhile to comment on the generalizability of the cost analysis across other parts of Africa. This is a very important topic in the HIV community as we look to scale up prevention efforts in areas with the greatest needs (i.e. SSA). This manuscript will be a wonderful addition to the current PrEP literature, especially from the implementation standpoint where costs play an important role. Thank you for the contribution to this extremely important public health and HIV/PrEP topic and I look forward to seeing the results of the main Ibumu study.
Author Response
Reviewer 2:
The authors have conducted a great nested study within the Ibumu clinical trial. There have been other prevention publications studying HIVST distribution from PWLHIV to their partners but this is the first to address the costs perspective on distribution strategies. The manuscript was well written. There was good supporting evidence and rationale for the study and the study design was thorough and well thought-out. The results were presented in a concise manner and with clarity. Appreciate the authors addressing the study limitations. It would be worthwhile to comment on the generalizability of the cost analysis across other parts of Africa. This is a very important topic in the HIV community as we look to scale up prevention efforts in areas with the greatest needs (i.e. SSA). This manuscript will be a wonderful addition to the current PrEP
literature, especially from the implementation standpoint where costs play an important role. Thank you for the contribution to this extremely important public health and HIV/PrEP topic and I look forward to seeing the results of the main Ibumu study.
Response: We thank the reviewer for the positive feedback regarding our analysis. We agree with the reviewer that we should emphasize the generalizability of our findings across other parts of SSA. We have added the following text to our discussion section on Line 430:
“Our costs are likely generalizable to other countries in SSA with similar HIV epidemics.”
Furthermore, we have edited the conclusion sentence of the abstract to read:
“Secondary distribution of HIVST from pregnant women can be implemented at reasonable cost to increase testing coverage among men in Uganda and similar settings in Africa.”